Corrected: Publisher correction

# Maturation of the gut microbiome and risk of asthma in childhood

Jakob Stokholm[1,2], Martin J. Blaser [2], Jonathan Thorsen[1], Morten A. Rasmussen[1,3,4], Johannes Waage[1], Rebecca K. Vinding[1], Ann-Marie M. Schoos[1], Asja Kunøe[1], Nadia R. Fink[1], Bo L. Chawes[1], Klaus Bønnelykke[1], Asker D. Brejnrod[5], Martin S. Mortensen [5], Waleed Abu Al-Soud[5], Søren J. Sørensen[5] & Hans Bisgaard[1]

The composition of the human gut microbiome matures within the first years of life. It has been hypothesized that microbial compositions in this period can cause immune dysregulations and potentially cause asthma. Here we show, by associating gut microbial composition from 16S rRNA gene amplicon sequencing during the first year of life with subsequent risk of asthma in 690 participants, that 1-year-old children with an immature microbial composition have an increased risk of asthma at age 5 years. This association is only apparent among children born to asthmatic mothers, suggesting that lacking microbial stimulation during the first year of life can trigger their inherited asthma risk. Conversely, adequate maturation of the gut microbiome in this period may protect these pre-disposed children.

[1] COPSAC, Copenhagen Prospective Studies on Asthma in Childhood, Herlev and Gentofte Hospital, University of Copenhagen, Ledreborg Alle 34, 2820 Gentofte, Denmark. [2] Departments of Medicine and Microbiology, and the Human Microbiome Program, New York University Langone Medical Center, 550 First Avenue, New York, NY 10016, USA. [3] Department of Biostatistics, Harvard School of Public Health, 655 Huntington Avenue, Boston, MA 02115, USA. [4] Section of Chemometrics and Analytical Technologies, Department of Food Science, University of Copenhagen, Rolighedsvej 30, 1958 Frederiksberg C, Denmark. [5] Section of Microbiology, Department of Biology, University of Copenhagen, Universitetsparken 15, 2100 Copenhagen, Denmark. Søren J. Sørensen and Hans Bisgaard contributed equally to this work. Correspondence and requests for materials should be addressed to S.J.S. (email: sjs@bio.ku.dk) or to H.B. (email: bisgaard@copsac.com)

The first years of our lives represent a period that is critical for susceptibility to environmental exposures. Here, lasting effects may be introduced to the developing immune system through complex host–environment interactions[1–3]. The human microbiome contains as many as $10^{14}$ bacteria, similar to the number of cells of an individual[4]. From birth, humans are continuously subjected to multiple exposures that influence microbiome ecology[1,2,5]. The composition of the gut microbiome matures within the first years of life[6–9] and the microbiome may have the ability to affect host immune maturation[10–12]. Perturbation of the microbiome during this critical period of development[13] may cause asthma, allergy, and other immunologic disorders[14–19]. Thus, the microbiome may be an important environmental factor that determines the transition from health to disease[1,3,20].

The neonatal microbial colonization patterns are greatly affected by mode of delivery[21,22] and use of intrapartum antibiotics[22,23], factors which may cause long-term microbial derangements[8,9], though the extent of these derangements differs between studies[22,24]. Birth by cesarean section is a recognized risk factor for asthma[25,26], as well as for other immune-mediated diseases in childhood[27]. Furthermore, antibiotic exposure in the first year of life has also been associated with increased asthma risk[28], again pointing to microbe-mediated mechanisms. Two studies have associated the gut microbial composition in the first year of life with early wheezy phenotypes[19,29] despite relying on questionnaire-based wheezy phenotypes and omitting microbial development over time.

The objective of this study is to analyze the nature of gut colonization patterns during the first year of life, and the associations of these patterns with the later risk of asthma among children from the Copenhagen Prospective Studies on Asthma in Childhood[2010] (COPSAC[2010]) birth cohort. We show that the gut microbiome, as assessed by β-diversity, relative abundances of genera and microbial maturation at age 1 year, is associated with asthma at age 5 years in the 690 children. This effect is only apparent in children born to asthmatic mothers, and especially characterizes an asthma phenotype also comprising allergic sensitization. This indicates that the development of the gut microbiome during the first year of life can impact the development of childhood asthma, and that adequate microbial maturation in this period may protect pre-disposed children.

## Results

**Baseline characteristics of the study cohort.** The COPSAC[2010] cohort has been followed prospectively with deep clinical phenotyping and structured interviews at 11 scheduled visits during the first 5 years of life, with asthma as primary outcome. The characteristics of the entire cohort of 700 children have been described in detail[30]. Of these children, 51% were boys, 57% had at least one older child in the home at birth, 22% were delivered by cesarean section, and 46% were treated with antibiotics during the first year of life. The mean maternal age at delivery was 32.2 years and 26% of the mothers had physician-diagnosed asthma.

**Microbial composition changes during first year of life.** The composition of the gut microbiome changes extensively in early life, with regard to diversity, complexity, and dominant bacterial taxa[5–9]. Therefore, we examined the compositional changes that occurred in the first year of life, before asthma onset. A total of 1696 fecal samples from 690 children arrived in the lab within 24 h of being produced and were characterized by 16S rRNA gene amplicon sequencing of the V4 region. With a median sequencing depth of 44,827 reads (interquartile range (IQR): 2358–78,208) increasing with age of sample, we identified 3651 unique operational taxonomic units (OTUs), demonstrating a median richness of 116 OTUs per sample, with the dominating genera *Bacteroides*, *Bifidobacterium*, and *Veillonella*. Alpha-diversity, assessed by the Shannon diversity index, was not different between 1 week (median (IQR), 2.0 (1.5–2.5)) and 1 month (1.9 (1.4–2.5)), but substantially increased at age 1 year (2.8 (2.4–3.3)), ($P < 0.001$). Similarly, the Chao1 index showed a minor decrease between 1 week (131 (96–175)) and 1 month (121 (78–168)) ($P < 0.001$) and a substantial increase at age 1 year (295 (203–366)) ($P < 0.001$). Although significant differences in the population structure (β-diversity), as determined by weighted UniFrac distances, were found between ages 1 week and 1 month ($F = 26.5$, $R^2 = 2.4\%$, $P < 0.001$), the greatest change in structure was between ages 1 month and 1 year ($F = 558.5$, $R^2 = 33.2\%$, $P < 0.001$, Fig. 1a). To evaluate which bacterial genera were involved in these observed temporal differences in α- and β-diversity, we examined the relative abundances of the most prevalent taxa. Each of the 20 most abundant genera changed significantly in relative abundance during the first year of life (Kruskal–Wallis test, all $P$-values < 0.01) with the largest differences observed from age 1 month to 1 year (Supplementary Fig. 1).

**Microbial community types and maturation are age-determined.** To further describe the compositional differences in the microbial populations, we applied a clustering method (Partitioning around medoids (PAM) clustering)[31,32] to separate all the fecal samples by their weighted UniFrac distances taking microbial phylogeny and abundances into account, but without including the time point in the computational model (Fig. 1b). This approach divided the samples into two distinct microbial community types (PAM clusters) (silhouette width, 0.30), largely representing the child's age at sampling, with PAM cluster 1 ($N = 1019$) mainly composed of the 1-week and 1-month samples and PAM cluster 2 ($N = 677$) chiefly composed of the 1-year samples (Fig. 1c). The five most discriminating indicator OTUs for each cluster were identified for PAM cluster 1 as *Enterobacteriaceae*, *Staphylococcus*, *Streptococcus*, *Bifidobacterium* and *Enterococcus*, and for PAM cluster 2 as *Faecalibacterium*, *Bacteroides(x3)*, and *Anaerostipes*. Using the Shannon diversity index, we identified higher α-diversity, as well as a microbiome dominated by the phylum Bacteroidetes in PAM cluster 2 compared to PAM cluster 1, which was dominated by Proteobacteria (Supplementary Fig. 2). As such, these PAM clusters likely represent the age-related maturation of the intestinal microbial populations[7,33,34]. To determine whether the PAM cluster transition at age 1 year was dependent on environmental factors, a wide range of exposures (including delivery method and antibiotics in the first year of life) were examined. Among these, only the presence of older children in the home from birth was significantly associated with the clusters; of the 34 children with a PAM cluster 1 composition at age 1 year, only 24% ($N = 8$) had older children in the home compared to 57% ($N = 334$) of the 589 children in PAM cluster 2 ($\chi^2$ test $P < 0.001$). The microbial populations by means of β-diversity at age 1 month was not associated with PAM cluster at age 1 year.

To further explore the microbial maturation process over time, we calculated microbiota-by-age z-scores (MAZ)[7] for all samples. The model is trained from the microbial composition on a data set with known sample age, and afterward predicts the microbial age for each sample based solely on the microbial composition. Microbiota age increased significantly for each adjacent time point (Wilcoxon test, $P$-values < 0.001) (Supplementary Fig. 3).

**Microbial population structure associates with later asthma.** Next, we sought to examine whether associations existed between

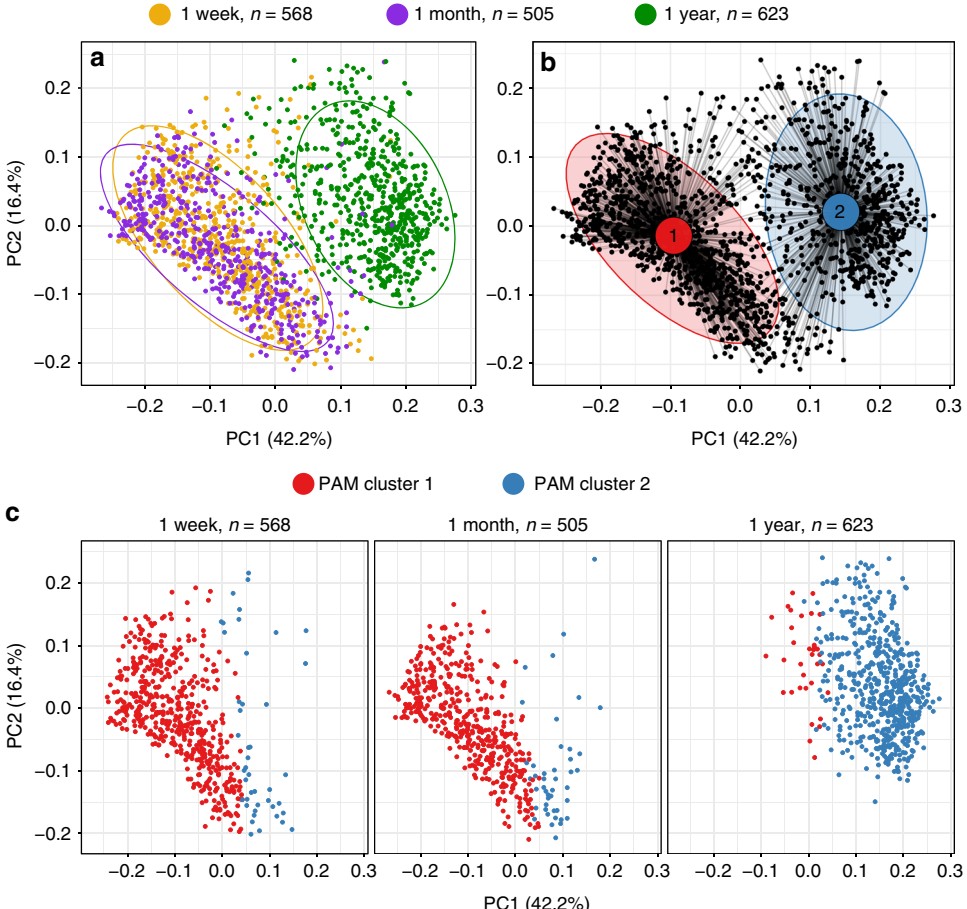

**Fig. 1** Microbial compositions change in the gut over the first year of life. PCoA plots of weighted UniFrac distances. **a** Separation of the composition by sampling time point. **b** Separation of the bacterial populations by Partitioning around medoids (PAM) clusters (optimal number of clusters is 2; silhouette width = 0.30). The five most distinctive indicator OTUs for each of the two clusters were: PAM cluster 1 (N = 1019): *Enterobacteriaceae*, *Staphylococcus*, *Streptococcus*, *Bifidobacterium* and *Enterococcus*, and PAM cluster 2 (N = 677): *Faecalibacterium*, *Bacteroides(x3)*, and *Anaerostipes*. Ellipses demonstrate the mean ± 2 SD in **a** and **b**. **c** Relationship between sampling time point and PAM cluster

the overall microbial composition at the different time points and the later development of asthma. Current asthma at age 5 years was used as the primary end-point, as it represents a strong persistent clinical phenotype; many children experience episodes of asthma-like symptoms in the first years of life, but outgrow their asthma symptoms before school-age[35]. Among the 648 children with follow-up to age 5 years, the prevalence of ongoing asthma at age 5 was 9% (N = 60). There were no significant associations between α-diversity (Shannon diversity and Chao1 indices) at any time point and asthma risk, also after adjustment for potential confounders (older siblings, duration of exclusive breastfeeding, hospitalization after birth, antibiotic use, and delivery mode). There were no significant associations between β-diversity at the two earliest time points; however, the microbial populations were significantly different at 1 year in children who had asthma at age 5 compared to non-asthmatics (F = 3.4, $R^2$ = 0.6%, P = 0.003). To further explore this finding, we stratified the samples at age 1 year according to maternal asthma status. The microbiome-asthma association was found only in the 147 children born to asthmatic mothers (F = 6.3, $R^2$ = 4.2%, P < 0.001) but not in children of non-asthmatic mothers (F = 0.7, $R^2$ = 0.2%, P = 0.65), demonstrating a significant interaction (P = 0.003) (Fig. 2). In a test of sensitivity, we adjusted these analyses for potential confounders (older siblings, duration of exclusive breastfeeding, hospitalization after birth, antibiotic use, and delivery mode). The significance levels remained essentially

unchanged (All children: F = 3.5, $R^2$ = 0.6%, P < 0.001; asthmatic mother: F = 6.2, $R^2$ = 4.1%, P < 0.001; non-asthmatic mother: F = 0.7, $R^2$ = 0.2%, P = 0.647). There were no differences in relation to asthmatic status of the father. Neither maternal nor paternal asthma status was associated with the α- or β-diversity at any time point (maternal asthma vs. β-diversity: Supplementary Fig. 4). However, that the microbial composition was not affected by maternal asthma status suggests that only susceptible children, exposed to inappropriate microbial stimulation during the first year of life, may express their inherited asthma risk. In a test of sensitivity, we also examined associations between population structure at age 1 year in children born by asthmatic mothers and asthma risk using other β-diversity indices that are not biased by the dominant taxa (unweighted UniFrac and Jensen–Shannon divergence), which yielded similar results (P < 0.001).

**Relative abundance at age 1 year associates with later asthma.** To understand whether the later asthma risk was affected by specific bacterial genera present earlier, we examined the relative abundances of the most common genera. For the 20 most abundant genera present in the 1-year samples, we observed increased risk of asthma at age 5 years associated with higher abundance of *Veillonella* (asthma vs. non-asthma; median relative abundance, 0.94 vs. 0.29%; Wilcoxon rank-sum test, P = 0.035) and with lower abundance of *Roseburia* (0.27 vs. 0.66%; P =

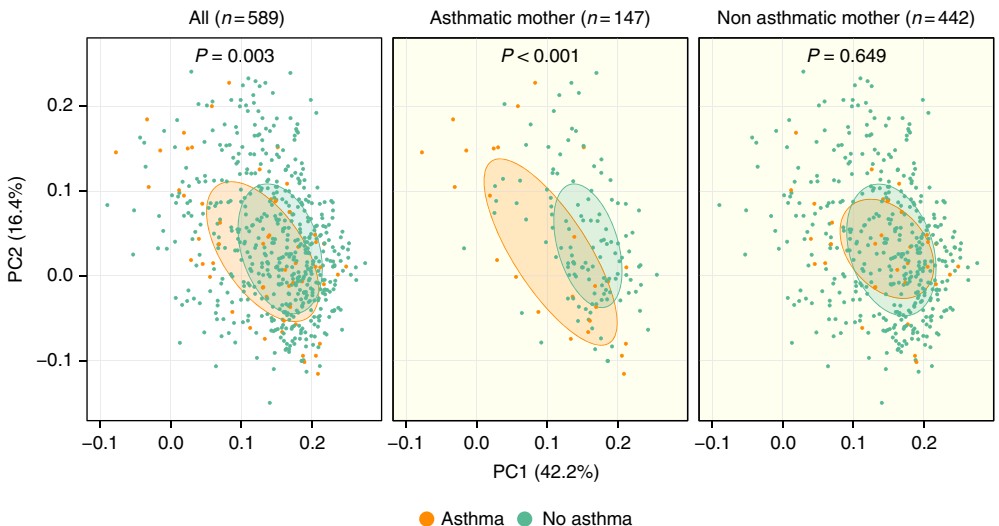

**Fig. 2** β-diversity in the 1-year fecal samples associates with later asthma. PCoA plots of weighted UniFrac distances. Microbial compositions are assessed in relation to a child's asthma status at age 5 years, and stratified by maternal asthma. *P*-values correspond to Adonis PERMANOVA tests. Ellipses demonstrate the mean ± 1 SD of children, who at age 5 years were asthmatic (orange) (*N* = 58) or non-asthmatic (green) (*N* = 531). Subsets include: asthmatic mother, asthmatic (orange) (*N* = 25), and non-asthmatic (green) (*N* = 122); non-asthmatic mother, asthmatic (orange) (*N* = 33), and non-asthmatic (green) (*N* = 409)

0.042), *Alistipes* (0.04 vs. 0.35%; *P* = 0.002), and *Flavonifractor* (0.05 vs. 0.07%; *P* = 0.002). If the child was born to an asthmatic mother, 8 of these 20 genera were significantly associated with their development of asthma, whereas there were no significant associations if the mother did not have asthma (Fig. 3). In children born to asthmatic mothers, asthma at age 5 years was negatively associated with relative abundance in the age 1-year sample of the genera *Faecalibacterium* (0.59 vs. 3.27%; *P* = 0.010), *Bifidobacterium* (0.47 vs. 2.27%; *P* = 0.006), *Roseburia* (0.01 vs. 0.76%; *P* < 0.001), *Alistipes* (0.01 vs. 0.39%; *P* = 0.003), *Lachnospiraceae incertae sedis* (0.05 vs. 0.19%; *P* = 0.018), *Ruminococcus* (<0.01 vs. 0.13%; *P* = 0.004) and *Dialister* (<0.01 vs. 0.15%; *P* = 0.007) and positively correlated only with *Veillonella* (1.41 vs. 0.23%; *P* = 0.039) (Fig. 3). The genera that were found in lower abundances at age 1 year among children who later became asthmatics have been considered as determinants of a healthy mature gut composition[36]. The majority of these genera were highly correlated at 1 year, while being negatively correlated with the genus *Veillonella* (Supplementary Fig. 5). None of the 20 most abundant genera at 1 week or 1 month were associated with asthma development. To further examine the associations between the microbial composition at age 1 year and asthma at age 5 in children born to asthmatic mothers, a cross-validated sparse PLS model was constructed to identify jointly contributing taxa at age 1 year that would predict later asthma in these children. This resulted in a one-component model based on the relative abundances of 60 genera (Supplementary Fig. 6). The model demonstrated a high predictive capacity for asthma (cross-validated AUC 0.76) and the many contributing taxa suggest a global delayed microbial maturation at age 1 year in children with asthma at age 5.

**Community types at age 1 year associates with later asthma.** To formally test whether the maturation of the microbiome was a determinant for asthma development, we used our microbial community types (PAM clusters) as indicators of microbial maturation. After excluding children without full 5-year follow-up, transient asthma and diagnosis before the 1-year sample (*N* = 105), the risk of developing persistent asthma if the child's microbiome remained in PAM cluster 1 at 1 year of age

(*N* = 28) was compared to children with transition to PAM cluster 2 (*N* = 490), and all analyses were adjusted for the presence of older children (Fig. 4). During the first 5 years of life, the risk of developing persistent asthma was increased (adjusted hazard ratio (aHR) 2.87 (1.25–6.55), *P* = 0.013) if the microbiome remained in PAM cluster 1 at age 1. This effect was driven solely by the 120 children born to asthmatic mothers (PAM1, *N* = 7 vs. PAM2, *N* = 113) (aHR 12.99 (4.17–40.51), *P* < 0.001), whereas there was no microbial effect on asthma development for the 398 children of non-asthmatic mothers (PAM1, *N* = 21 vs. PAM2, *N* = 377) (aHR 0.56 (0.07–4.16), *P* = 0.57), demonstrating significant interaction, *P* = 0.011. In a test of sensitivity, we adjusted these analyses further by including other potential confounders (older siblings, duration of exclusive breastfeeding, hospitalization after birth, antibiotic use, and delivery mode). All estimates and significance levels remained essentially unchanged (all children: aHR 2.91 (1.25–6.79), *P* = 0.013; asthmatic mother: aHR 10.92 (3.44 –34.67), *P* < 0.001; non-asthmatic mother: aHR 0.56 (0.07–4.23), *P* = 0.572). We found no associations of PAM clusters with the transient asthma phenotype. These analyses provide further evidence that having the more immature gut microbial composition at age 1 year may be a trigger for asthma development in susceptible children.

**Low 1-year microbial maturity associates with later asthma.** We then applied another validated method for determining the early life maturation of the intestinal microbial populations using the MAZ score. The microbial maturity (predicted microbiota age −median microbiota age) and MAZ (microbial maturity/SD) were used as metrics[7,33,34]. We sought to determine whether a lower MAZ score at age 1 year was associated with later asthma. During the first 5 years of life, the risk of developing persistent asthma was increased with low maturity (below median) in MAZ at 1 year (low maturity, *N* = 257 vs. high maturity, *N* = 261) (HR 1.77 (1.02–3.07), *P* = 0.043). Low microbial maturity was only associated with later asthma in children born to asthmatic mothers (low maturity, *N* = 63 vs. high maturity, *N* = 57) (HR 6.53 (1.93–22.06), *P* = 0.003), and not in children of non-asthmatic mothers (low maturity, *N* = 194 vs. high maturity, *N* = 204) (HR 0.92 (0.46–1.84), *P* = 0.81), demonstrating significant

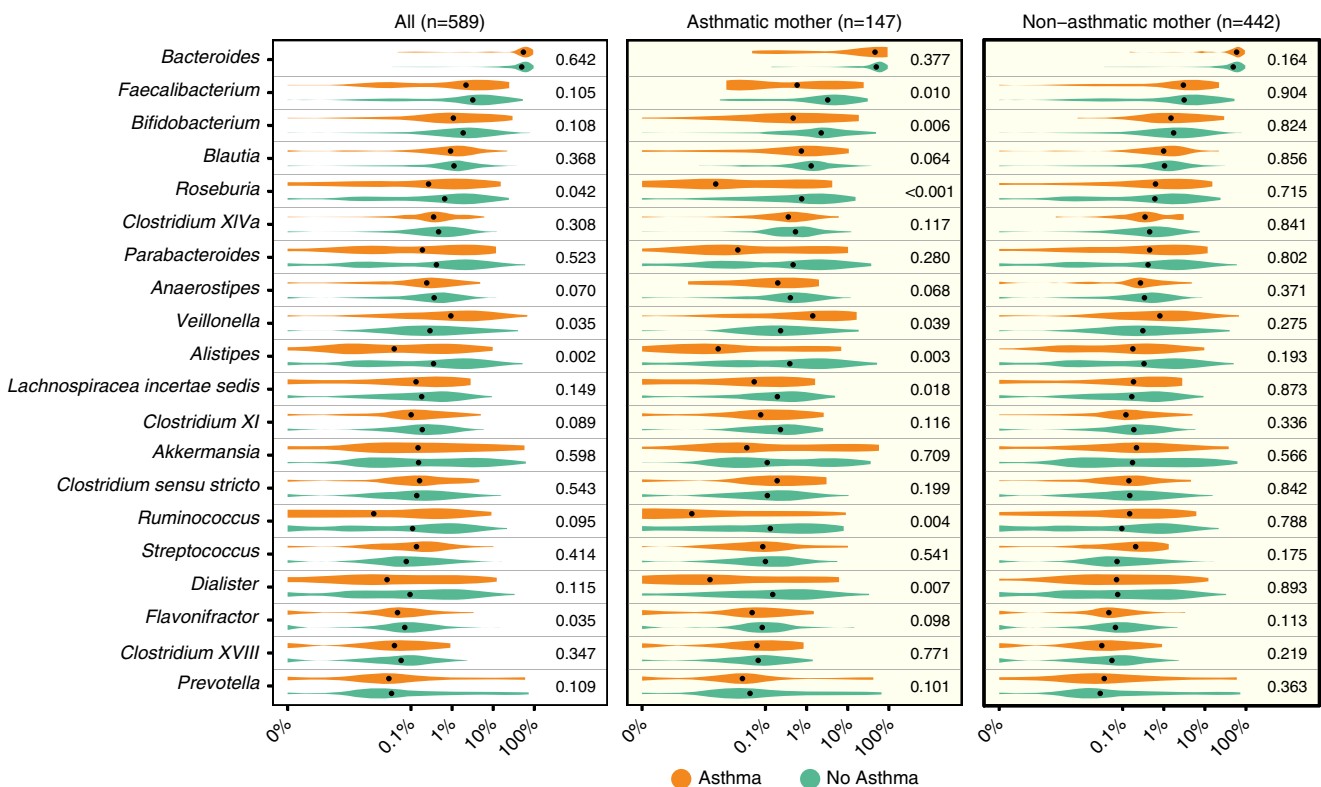

**Fig. 3** Relative abundances in the 1-year fecal samples associate with later asthma. Comparison among the 20 most abundant bacterial genera. Relative abundance of each genus is shown with respect to asthma at age 5 years in all children, and stratified by maternal asthma. *P*-values correspond to Wilcoxon rank-sum tests of the relative abundances, with significant values (*P* < 0.05) bolded. FDR limits were calculated for the comparisons: Bonferroni (all: *P* < 0.0025), Benjamini & Hochberg (all children: *P* < 0.0024, asthmatic mother: *P* < 0.0102, non-asthmatic mother: *P* < 0.05). A pseudocount (+1e −06) was added to all abundances for the log-scale presentation. The black dots indicate median values and the abundances are colored according to the asthmatic (orange) (*N* = 58) or non-asthmatic (green) (*N* = 531) status of the child at age 5 years

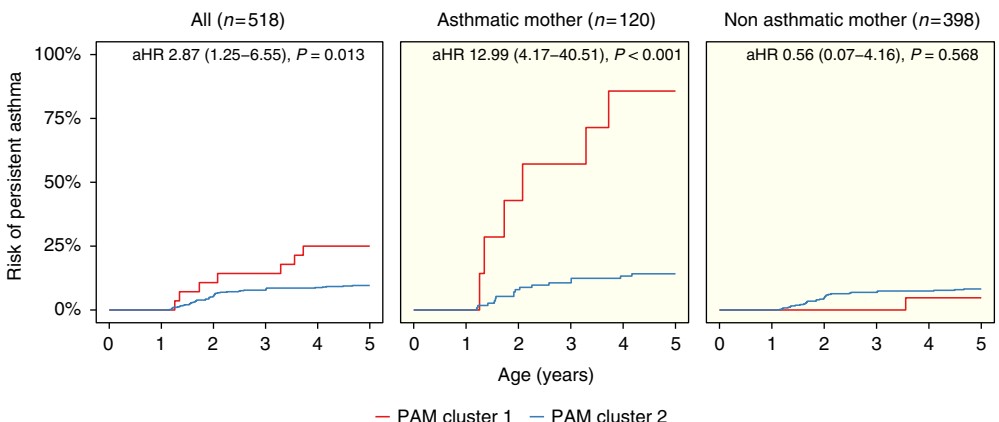

**Fig. 4** Kaplan–Meier plots illustrate risk of asthma in the first 5 years of life according to Partitioning Around Medoids (PAM) cluster at 1 year. Comparisons shown of all children and stratified by maternal asthma, demonstrating interaction between PAM cluster and maternal asthma (*P* = 0.01). Associations are quantified by Cox proportional hazards regression, adjusted for presence of older children in the home (aHR, adjusted Hazard Ratio). Children with a transient asthmatic phenotype (remission before age 5 years) are excluded. Curves are colored according to cluster; All: PAM cluster 1 (red) (*N* = 28), PAM cluster 2 (blue) (*N* = 490). Subsets include: asthmatic mother, PAM cluster 1 (red) (*N* = 7), PAM cluster 2 (blue) (*N* = 113); non-asthmatic mother: PAM cluster 1 (red) (*N* = 21), PAM cluster 2 (blue) (*N* = 377)

interaction, *P* = 0.006 (Supplementary Fig. 7). In total, adequate maturation of the gut microbiome in this period appears critical for healthy (non-asthmatic) development and may protect children with pre-dispositions.

**Low microbial maturity associates with number of asthma-like episodes**. We then examined a secondary asthmatic end-point, the total number of asthma-like episodes in the first 3 years of life, which is a marker of disease burden. Children with microbiome

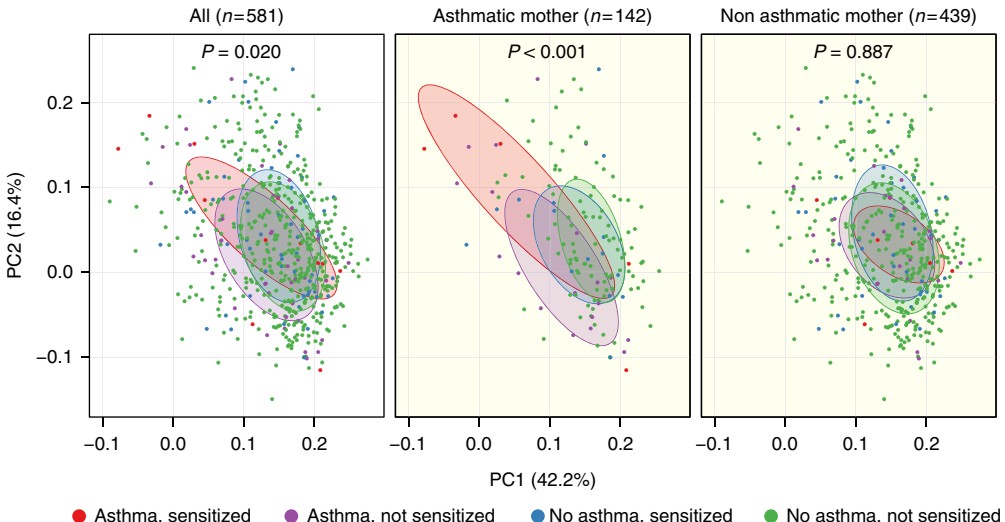

**Fig. 5** β-diversity at 1 year associates with asthma and allergic sensitization. PCoA plots of weighted UniFrac distances. Microbial compositions are assessed in relation to a child's asthma status at age 5 years and allergic sensitization before age 18 months, and stratified by maternal asthma. *P*-values correspond to Adonis PERMANOVA tests. Ellipses demonstrate the mean ± 1 SD of children, who at age 5 years were: asthmatic and sensitized (red) (*N* = 12), asthmatic and not sensitized (purple) (*N* = 45), non-asthmatic and sensitized (blue) (*N* = 59), or non-asthmatic and not sensitized (green) (*N* = 465). Subsets include: asthmatic mother, asthmatic and sensitized (red) (*N* = 5), asthmatic and not sensitized (purple) (*N* = 19), non-asthmatic and sensitized (blue) (*N* = 15), and non-asthmatic and not sensitized (green) (*N* = 103); non-asthmatic mother: asthmatic and sensitized (red) (*N* = 7), asthmatic and not sensitized (purple) (*N* = 26), non-asthmatic and sensitized (blue) (*N* = 44), and non-asthmatic and not sensitized (green) (*N* = 362)

composition in PAM cluster 1 at 1 year experienced more episodes of troublesome lung symptoms compared with PAM cluster 2 (median (IQR); PAM1 vs. PAM2, 6 (4–18) vs. 5 (2–10); incidence risk ratio (IRR) 1.54 (1.12–2.12), *P* = 0.008). This was more pronounced in children born to asthmatic mothers (interaction *P* = 0.040, median (IQR); PAM1 vs. PAM2, 19 (12–27) vs. 6 (3–12); IRR 2.35 (1.48-3.74), *P* < 0.001). Likewise, lower microbial maturity as assessed by MAZ score at age 1 year was associated with more episodes (IRR 1.13 (1.06–1.20), *P* < 0.001), again with an enhanced effect in children of asthmatic mothers IRR 1.26 (1.05–2.91), *P* = 0.016. Thus, consistent with the findings for asthma development, low maturity of the microbiome at age 1 year was also associated with more asthma-like episodes.

**The associations remain robust at higher sequencing depth**. To test the robustness of these findings, the analyses were stratified by sequencing depth to only include samples with more than 10,000 reads. The PAM clustering, MAZ, and association results with asthma and episodes of troublesome lung symptoms all were robust with reference to this more stringent inclusion criterion. Furthermore, we found no significant differences in sequencing depth at 1 year between children, who did or did not develop asthma (median (IQR); asthma vs. non-asthma, 54,260 (12,018–74,971) vs. 51,940 (12,010–73,050), *P* = 0.8); adjustment of the results for sequencing depth resulted in essentially no difference.

**Microbial populations describe asthma with sensitization**. To explore whether the results were characterizing a specific asthmatic phenotype, the children were further subdivided by allergic sensitization in childhood. The microbial populations by means of β-diversity at age 1 year were significantly different whether the child had asthma or not with or without sensitization (four categories of comparison) (*F* = 1.7, $R^2$ = 0.9%, *P* = 0.020). Again, this was mainly apparent among children born to asthmatic mothers (*F* = 2.9, $R^2$ = 5.9%, *P* < 0.001) with no significant effects among children born to non-asthmatic mothers *F* = 0.7, $R^2$ = 0.5%, (*P* = 0.887). Especially the phenotype of asthma with

allergic sensitization was associated with skewed β-diversity (Fig. 5).

## Discussion

With the primary objective of making accurate asthma diagnoses and comprehensive phenotyping, we found that the gut microbiome at age 1 year was associated with asthma at age 5 in a prospective cohort of 700 children. Parallel observations were apparent for overall β-diversity, relative abundances of genera, and maturation by means of PAM clustering and MAZ. The effect was only apparent in children born to asthmatic mothers, but the composition and maturation of the gut microbiome in early life was not affected by maternal asthma status. Because of the observational design of the study, we cannot unravel causality or directionality, but our findings indicate that children born to asthmatic mothers harbor a susceptibility to influences from the gut microbiome and that adequate maturation in this period may protect these pre-disposed children.

The main study strength is the design of the COPSAC<sub>2010</sub> birth cohort, which is a replication and extension of the COPSAC<sub>2000</sub> cohort[37] with regular longitudinal clinical follow-up, meticulous daily symptom recording, and acute care visits to the clinical research site, which has a long record of performing clinical follow-up in cohort studies[30,37]. All diagnoses and treatments of asthma were performed solely by COPSAC physicians[30]. This ensured highly specific end-points and accurate assessment of time of disease onset. Importantly, this allowed for discrimination between persistent and transient asthma phenotypes. Asthma at age 5 years is a robust phenotype compared with wheezing in the first few years of life, which for most children merely describes a transient phenotype, especially when accompanied by increased early life severity and sensitization[35,38,39]. Collecting fecal samples longitudinally from age 1 week allowed for description of the development of the microbial populations during the first year of life.

Our study is by design observational, which limits assessment of causal relationships. However, the number of participants allows for stratification of the data by physician-diagnosed

maternal asthma, which suggests potential mechanisms. Our observation also provides an initial indication of which children might benefit from microbial manipulation in the first year of life. Although the 16S rRNA gene amplicon sequencing technique provides a thorough description of the overall microbial populations, with sequencing of only the V4 region, the method only allows for robust classification of the individual taxa at the genus level. This limits our interpretation of the results, as we cannot describe the direct effect of specific species or strains. Furthermore, we cannot rule out that other gut microbiome components, which have not been characterized in the study (e.g., gut viral and/or fungal colonization), could be partly or fully responsible for some of the observed effects.

Our results suggest that the development of persistent asthma in childhood may be mediated through a skewed maturation of the gut microbiome in the first year of life, predominantly in children born to asthmatic mothers. Since the hygiene hypothesis was first described[40], multiple studies have linked indirect putative causes of microbial alterations in the first years of life with later risk of immune-mediated diseases[3]. Laboratory models have also provided proof of the microbiome's ability to modulate immune functions of the host[2,13,41]. Delivery mode has been studied intensively and the associations of cesarean section-birth with asthma[25,26] and other chronic childhood disorders[27] have been largely consistent. Furthermore, delivery by cesarean section greatly influences the earliest microbiome and its development[5,21,22,42]. Antibiotics in the first year of life have also been confirmed both to perturb the gut microbiome[8,9] as well as being associated with asthma risk[28]. However, these only represent indirect associations between microbial alterations and later disease risk, whereas we show direct and robust associations between the overall β-diversity, taxon relative abundances, and microbial maturation and risk of asthma at age 5.

Of the 20 most abundant genera in the samples of the children at age 1 year, we reported eight associated with asthma at age 5 in children born to asthmatic mothers. The most abundant Firmicutes families in the healthy adult human gut are *Lachnospiraceae* (including *Lachnospiraceae incertae sedis* and *Roseburia*) and *Ruminococcaceae* (including *Ruminococcus* and *Faecalibacterium*)[36]. The lower relative abundances of these genera in children who later become asthmatic could indicate overall delayed microbial maturation[7]. A study[19] from the Canadian Healthy Infant Longitudinal Development (CHILD) cohort identified associations between lower *Faecalibacterium* and *Lachnospira* at age 3 months and an early allergic wheezy phenotype at age 1 year, consistent with our findings. However, opposite to our study, they identified *Veillonella* as being protective. This dichotomy could be explained by differences in the disease phenotypes studied, since early life wheeze is a heterogeneous group[35] not directly comparable with the 5-year asthma phenotype we studied. However, the protective association of *Veillonella* in the CHILD study was not apparent in the 1-year samples, which is similar to our findings. In the genus *Faecalibacterium*, the only known species is *F. prausnitzii*, which is a butyrate-producing bacterium, considered to have anti-inflammatory activities[43]. Conversely, lack of *F. prausnitzii* is a marker for inflammatory bowel disease[44], obesity, and diabetes. We found *Faecalibacterium* as the most specific indicator for the more mature PAM cluster 2 and in lower abundance among children who later became asthmatic. This, and the association with the age-related microbiota score (MAZ), are consistent with a potential causal role, and could suggest a mechanism of delayed maturity influencing immune development in susceptible children. For decades, *Bifidobacterium* has been recognized as a potential protector against allergic disease[45] and lack of *Ruminococcus* in earliest life has also been associated with allergic

sensitization[46]. The lack of association between bacterial taxa at 1 week and 1 month and later asthma points toward microbial alterations going on between 1 month and 1 year. A deviation might have been observed already at 3 months as suggested in the CHILD study[19]. Furthermore, a skewed microbial composition at age 1 year was mainly associated with an allergic asthmatic phenotype, which is also in line with findings from the CHILD study[19]. The univariate genus-level relative abundance results were further expanded in a multivariate supervised model, again pointing toward differences in the entire bacterial community structure. For children born to asthmatic mothers, we demonstrated a cross-validated AUC of 0.76 for predicting asthma at age 5 years from the microbial composition at 1 year. In comparison, a large study using genetic risk scores of the highest-ranked asthma risk alleles demonstrated an AUC of 0.54–0.57 for prediction of school-age asthma, and an AUC of 0.59 for preschool persistent wheeze[47].

Maternal asthma status did not affect the microbial populations of the children and therefore did not confound our results. Maternal asthma was however, a key effect modifier between the microbiome and asthma risk in our study, which points to susceptibility to host–microbial interactions specifically for these children. Such susceptibility could arise from an inborn immune deviation determined by maternal asthma status, as we have reported in this cohort[48]. Stronger heritability of maternal over paternal asthma has been described[49], consistent with our findings, which points towards mechanisms beyond strict genetic effects.

Our results suggest that the lack of maturation of the gut microbiome in the first year of life is the critical determinant for increased asthma risk as illustrated by the microbial community type analyses. Having older siblings was the only significant determinant for belonging in the more mature PAM cluster 2 at age 1 year, and is consistent with a microbial mechanism for the original hygiene hypothesis[40]. We speculate that microbial transfer from the older child may advance maturation of the gut composition and, through appropriate stimulation of the developing immune system, cause protection from asthma in otherwise susceptible children.

Probiotic intervention trials in pregnancy and in early life have so far failed to prove effective in reducing asthma risk[50]. However, this could be due to species selection or inadequate (single or few species) compositions of the probiotics. It can be speculated that the global microbial population and delayed maturation in the first year of life matters more than individual taxa. Our results suggest possible beneficial effects of at least seven of the most abundant genera among the children born to asthmatic mothers. Because our results are observational and most of the taxa associated with lower asthma risk demonstrate collinearity, we are not able to pinpoint specific causal taxa; however, the multivariate model illustrates many jointly contributing taxa and not a single risk taxon, consistent with a global effect of the microbiota. Therefore, contributing bacterial strains must be identified, potential mechanisms validated, and immunological features determined in future experimental models. We speculate that a mixture of the strains to be identified could form the basis for a next-generation probiotic that could be tested in a randomized controlled trial to children at risk.

Our results suggest that the maturation of the gut microbiome during the first year of life has an important role in the development of childhood asthma, especially in children born to asthmatic mothers, and that older children in the home may help advance the maturation. Our results suggest potential beneficial effects of specific microbial supplementation in the first year of life for children at high risk for developing asthma.

## Methods

**Ethics statement.** The study was conducted in accordance with the guiding principles of the Declaration of Helsinki and was approved by The National Committee on Health Research Ethics (H-B-2008-093) and the Danish Data Protection Agency (2015-41-3696). Both parents gave written informed consent before enrollment.

**Study population.** The COPSAC_2010 cohort is a population-based birth cohort of 700 children recruited in pregnancy and has been followed prospectively at the COPSAC research unit with deep clinical phenotyping at 11 scheduled visits during the first 5 years of life. Of these 690 had a fecal sample characterized. The study pediatricians collected all information during these clinical visits scheduled at 1 week, 1, 3, 6, 12, 18, 24, 30, and 36 months, and yearly thereafter. Additional acute care visits were arranged whenever the children experienced lung or skin symptoms[30]. The symptom burden between visits was captured with daily diary cards monitoring: significant troublesome lung symptoms including components of cough, wheeze, and dyspnea; skin symptoms; and respiratory infections. The study pediatricians were the only physicians responsible for diagnosis and treatment of asthma, allergy, and eczema adhering to predefined algorithms[30].

**Primary end-point.** Asthma was diagnosed based on a previously detailed quantitative symptom algorithm requiring all of the following criteria[51–53]: (1) verified diary recordings of five episodes of troublesome lung symptoms within 6 months, each lasting at least 3 consecutive days; (2) symptoms typical of asthma including exercise-induced symptoms, prolonged nocturnal cough, and/or persistent cough outside of common colds; (3) need for intermittent rescue use of inhaled β2-agonist; and (4) response to a 3-month course of inhaled corticosteroids and relapse upon ending treatment[51]. Remission was defined by 12 months without relapse upon cessation of inhaled corticosteroid treatment. For analyses, we used the cross-sectional ongoing asthma diagnosis at age 5 years and the time to disease onset of either transient or persistent asthma (ongoing diagnosis at age 5 years).

**Secondary end-point.** Episodes of troublesome lung symptoms included the number of episodes lasting three or more consecutive days in the first 3 years of life, with symptoms including cough, wheeze, and/or dyspnea severely affecting the well-being of the child.

**Covariates.** Information on physician-diagnosed asthma in the mother and father (yes/no), maternal age at birth, smoking during pregnancy (yes/no), education (3 levels), household income (3 levels), antibiotics in pregnancy (yes/no, for any treatment), preeclampsia (yes/no), gestational diabetes (yes/no), delivery type (three levels: vaginal delivery, emergency cesarean section, or elective cesarean section), gestational age, hospitalization of the child after birth (yes/no), antibiotics to the child (yes/no, for any treatment before age 1 year), household pets (yes/no, for cat and/or dog), older children in the home (yes/no), and duration of exclusive and total breastfeeding period was obtained during the scheduled visits to the COPSAC clinic. Information on intrapartum antibiotics and antibiotic use during pregnancy and childhood was validated against national registries[54].

Allergic sensitization was determined at 6 and 18 months of age as any skin prick test (SPT) ≥2 mm (ALK-Abello, Horsholm, Denmark) and by specific IgE (sIgE) ≥0.35 kUa/L against milk, egg, dog, or cat (ImmunoCAP; Thermo Fischer Scientific, Allerod, Denmark)[30]. Children classified as "not sensitized" were both SPT and specific IgE negative for all tested allergens.

**Sample collection.** Fecal samples were collected 1 week, 1 month, and 1 year after birth, either at the research clinic or by the parents at home using detailed instructions. Each sample was mixed on arrival with 1 mL of 10% vol/vol glycerol broth (Statens Serum Institut, Copenhagen, Denmark) and frozen at −80 °C.

**DNA extraction.** Genomic DNA was extracted from the infants' samples using the PowerMag® Soil DNA Isolation Kit optimized for epMotion® (MO-BIO Laboratories, Inc., Carlsberg, CA, USA) using the epMotion® robotic platform model 5075 (Eppendorf) according to the manufacturer's protocol with the following alterations to the workflow: 150–250 µL of the samples were added to the 96-well bead plate containing 750 µL bead/RNase A Solution and 60 µL lysis solution. Centrifugation steps were performed at 3220xRCF for 9min. Removal of enzymatic inhibitors and DNA purification was performed as described by the manufacturer. Finally, the DNA was eluted with 100 µL Tris buffer (10 mM, pH 7.5). DNA concentrations were determined using the Quant-iT™ PicoGreen® quantification system (Life Technologies, CA, USA). Extracted DNA was stored at −20 °C.

**16S gene amplicon sequencing.** The 16S rRNA gene amplification procedure was divided into two PCR steps. In the first PCR reaction, the hypervariable V4 region of the 16S rRNA gene was amplified using the modified broad range primers 515F (5′-GTGCCAGCMGCCGCGGTAA-3′) and 806R (5′-GGAC-TACHVGGGTWTCTAAT-3′)[55–57]. Amplification was performed in 96-well microtiter plates with a reaction mixture consisting of 1× AccuPrime PCR Buffer II, 0.6 U AccuPrime Taq DNA Polymerase (Invitrogen, Life technologies, CA,

USA), 0.5 µM primer 515 F, 0.5 µM primer 806 R, and 2 µL template DNA, giving a total volume of 20 µL per sample. Reactions were run in a 2720 thermal cycler (Applied Biosystems®, Life Technologies, CA, USA) according to the following cycling program: 2 min of denaturation at 94 °C, followed by 30 cycles of 20 s at 94 °C (denaturing), 30 s at 56 °C (annealing), and 40 s at 68 °C (elongation), with a final extension at 68 °C for 5 min. For each plate, a negative template-free control and a positive control containing 2 µL DNA from a known bacterial mock community (1 ng/µL; HM-782D, BEI Resources, VA, USA) were included. The PCR products were quantified using the Quant-iT™ PicoGreen® quantification system (Life Technologies) and samples with a concentration above 6 ng/µL were diluted to ~3–6 ng/µL prior to further analysis.

Sequencing primers and adaptors were added to the amplicon products in the second PCR step as follows: 2 µL of the diluted amplicons were mixed with a reaction solution consisting of 1× AccuPrime PCR Buffer II, 0.6 U AccuPrime Taq DNA Polymerase (Invitrogen, Life Technologies) and 0.5 µM fusion forward and 0.5 µM fusion reverse primer (total volume 20 µL). The PCR was run according to the cycling program above except with a reduced cycling number of 15. The amplification products were purified with Agencourt AMPure XP Beads (Beckman Coulter Genomics, MA, USA) according to the manufacturer's specifications using 0.7× volume beads and quantified as described above. Equimolar amounts of the amplification products were pooled together in a single tube. The pooled DNA samples were concentrated using the DNA Clean & Concentrator™–5 Kit (Zymo Research, Irvine, CA, USA) according to the manufacturer's instructions. The concentration of the pooled libraries was determined using the Quant-iT™ High-Sensitivity DNA Assay Kit (Life Technologies) following the specifications of the manufacturer. Amplicon sequencing was performed on the Illumina MiSeq System (Illumina Inc., CA, USA). For each run, a 1.0% PhiX internal control was included. All reagents used were from the MiSeq Reagent Kits v2 (Illumina Inc.). Automated cluster generation and 2 × 250 bp paired-end sequencing with dual-index reads were performed. The sequencing output was generated as demultiplexed fastq-files for downstream analysis. Up to 192 samples were sequenced per run.

**Data processing.** Fastq-files demultiplexed by the MiSeq Controller Software (Illumina Inc.) were trimmed for amplification primers, diversity spacers, and sequencing adapters using biopieces (www.biopieces.org), mate-paired and quality filtered (USEARCH v7.0.1090)[58]. UPARSE[59,60] was used for OTU clustering as recommended, in particular removing singletons after dereplication. Chimera checking was performed with UCHIME[61] against Mothurs supplied version of the RDP9 PDS database. Representative sequences were classified at 0.8 confidence threshold (Mothur v1.25.0 wang function)[62]. FastTree[63] in nucleotide mode and Mothurs align.seq function[64] were used to construct a phylogenetic tree. Alignments were built with reference to the 2013 version of Greengenes[65].

**Statistics and data analysis.** All data analysis was performed in the statistical software package R version 3.3.0, with the package phyloseq[66] to handle the microbiome data. The Shannon diversity index and the Chao1 index were used as measures of the within individual diversity (α-diversity), and the between individual diversity metrics (β-diversity) were computed as weighted UniFrac[67] distances after adding a pseudocount (+1) to all OTU counts and log-transforming the OTU tables. For sensitivity analyses unweighted UniFrac and Jensen–Shannon divergence were used as other metrics of β-diversity. Differences in α-diversity between groups were compared using Wilcoxon rank-sum test and differences in β-diversity were visualized with principal coordinates analysis (PCoA) plots, in which the PCoA ordinations were calculated based on all the samples visualized in the specific plot and subsequently stratified. β-diversity was tested for inference using permutational multivariate analysis of variance (PERMANOVA) (Adonis from the package vegan[68,69] with 999 permutations). In the interaction analyses, we included an interaction term (e.g., asthma * asthma_status_mother). Differences in relative abundances at the genus level were analyzed using Wilcoxon rank-sum (2 levels) or Kruskal–Wallis Test (>2 levels). We used sparse partial least squares (PLS)[70] modeling of asthma at age 5 years after filtering genera (prevalence >5% of children, >0.01% mean relative abundance) and log-transforming relative abundances, using half the lowest nonzero value as a pseudocount. We selected the optimum number of input variables using repeated 10-fold cross-validation of the area under the curve (AUC) statistic to avoid overfitting. The final model was chosen by the highest median AUC value. The predicted values of left out folds were combined to a PLS score. Loading of each genus in the model was in the figure drawn with standard deviations across repeats as error bars, ranked by relative abundance and colored by phylum. Clustering analysis was performed using partitioning around medoids (PAM) clustering[31] from the package cluster[32]. The Silhouette index[71] calculated from weighted UniFrac distances was used for determining the optimal number of clusters in the data. Indicator OTUs were identified using the function multipatt from the package indicspecies[72]. Microbiota-by-age z-scores (MAZ)[7] were calculated with non-asthmatics serving as the training set, with calculation of: microbial maturity (MM) = (predicted microbiota age − median microbiota age); and MAZ = (MM/SD) of predicted microbiota age as metrics. Associations between PAM clusters and MAZ and asthma were assessed by Kaplan–Meier analyses and quantified by Cox proportional hazards regression (P-values correspond to Wald tests). The children were retained in the analysis from birth until age of asthma

diagnosis, drop-out, or age 5 years, whichever came first. Children with an asthma diagnosis before sampling was excluded. The heatmap was constructed using the package "pheatmap" and illustrates Spearman correlations between taxa, filtered by >10% genus presence and >$10^{-3}$ mean relative abundance. Number of episodes of troublesome lung symptoms was analyzed by a generalized estimating equation Poisson regression model. Covariate analyses were conducted using $\chi^2$ tests for categorical variables and $t$-tests or Wilcoxon rank-sum test for continuous variables. A significance level of 0.05 was used in all analyses.

**Data availability**. The 16S sequences have been deposited in the Sequence Read Archive (SRA) repository with the accession number PRJNA417357. All other relevant data are available from the authors upon reasonable requests.

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

## Acknowledgements

We express our deepest gratitude to the children and families of the COPSAC$_{2010}$ cohort study for all their support and commitment. We acknowledge and appreciate the unique efforts of the COPSAC research team. J.S. has received funding from the Alfred Benzon Foundation, the Thrasher Research Fund and the Danish Council for Independent Research. COPSAC is funded by private and public research funds all listed on www.copsac.com. The Lundbeck Foundation; The Danish Ministry of Health; Danish Council for Strategic Research and The Capital Region Research Foundation have provided core support for COPSAC. M.B. was supported in part by R01 DK090989 from the National Institutes of Health and by the Ziff Family and C&D funds. All authors have agreed that the accuracy and integrity of any part of the work has been appropriately investigated and resolved and all have approved the final version of the manuscript. The corresponding author had full access to the data and had final responsibility for the decision to submit for publication.

## Author contributions

The guarantor of the study is H.B., from conception and design to conduct of the study and acquisition of data, data analysis, and interpretation of data. All co-authors have contributed substantially to the analyses and interpretation of the data, and have provided important intellectual input. J.S. was responsible for acquisition, analysis, and interpretation of data and has written the first draft of the manuscript. J.T. contributed with analysis and interpretation. A.D.B. wrote the bioinformatics pipeline.

## Additional information

**Competing interests:** MJB discloses his participation on the Scientific Advisory Board of Commense, Inc. The other authors declare no competing financial interests.

