## [Peer Review File · Nature Communications]

PEER REVIEW FILE

Reviewers' comments:

Reviewer #1 (neonate gut microbiome, asthma)(Remarks to the Author):

This version of the manuscript is improved and maintains the core message that a certain 'immature' microbiome can predispose children, born from an asthmatic mother, to asthma at 5 years. This is the novel contribution the report brings to the field.

The authors should include a short, referenced, discussion that compares the effect size underlying their primary conclusions (e.g. maternal asthma-immature microbiota-asthma at 5y) with other asthma risk factors, e.g. environment, antibiotics, recurrent viral infections. It is important that this comparison is discussed in order for readers to be able to put the significance of the current report into context.

Minor comment. Table 1, which is the PICRUS dataset, has not been removed from the submitted files.

Reviewer #2 (gut microbiome)(Remarks to the Author):

The manuscript submitted by Stockholm et al. describes the microbiota at several time-points in early life in relation to asthma development. The study is conducted within the context of the longitudinal COPSAC2010 birth cohort, making it one of the few studies that moves beyond cross-sectional analyses. Moreover, in this revised version of the manuscript the authors have taken into account the relevant metadata that could have potentially confounded the results. Also other comments by all three previous reviewers have in my opinion in general be carefully addressed. There are only a few issues that still remain that I think should be addressed:

1. Median sequencing depth is insufficient to provide insight in the quality of the data. If the variation in sequencing depth is large, this could have limited the power of the study to reveal

additional associations in particular regarding alpha diversity. This aspect is not solved by testing for differences in sequencing depth between groups or adjusting for sequencing depth (although this is a relevant addition the authors made to rule out confounding).

Please provide range in sequencing depth along with median.

2. Although the authors have now incorporated the potential confounders in the analyses related to microbial population structure and PAM clustering, it is unclear why the authors chose not to do the same for the alpha-diversity analyses. Associations can also be masked due to confounding, so a lack of association is not an argument not to adjust for covariables. I would recommend the minor effort to adjust alpha diversity analyses as well.

3. The authors indicated in their response that they deliberately focused the manuscript on the entire community structure. However, the point is that the weighted Unifrac that has been used as the only metric for describing population structure and PAM clustering etc., is heavily biased towards the dominant bacterial taxa (see PMID 22711789). By only focussing on this metric as a beta-diversity indice, the aim to explore the entire community structure in association to asthma development is therefore not fully explored. I would recommend to either confirm or extend major findings with an additional indice that is not skewed towards the dominant taxa or address the limitations of using the weighted Unifrac only in the discussion of the manuscript.

Reviewer #3 (human microbiome)(Remarks to the Author):

Authors have attempted to respond to first round of reviews

My concerns remain - incremental advance, 16S only data set of limited value and follow up samples are critical

Reviewer #4 (functional genomics, biostat)(Remarks to the Author):

In this paper, Stokholm and colleagues characterized the microbial composition of fecal samples from children in the COPSAC child cohort at three different ages (1 week, 1 month, and 1 year) using 16S rRNA gene sequencing. The goal of this prospective study was to evaluate the association between the gut microbiota during the first year of life and asthma diagnosis in later year. The current manuscript has been revised from the original submission. The authors have addressed every issue raised by the three previous reviewers. I am in agreement with most of the critiques raised by the three previous reviewers. My review is primarily focused on whether the responses from the authors are adequate and the revised version has been improved. I only have minor points raised from my own reading of the manuscript. Overall, I felt that the cohort is a unique with both microbiome data as well as deep clinical information and the study is well

designed to take advantage of the prospective cohort with sufficient power. The finding is interesting and make sense. However, given the current analysis result, the difference appears to be weak, which cast doubt whether the finding would have any clinical implication.

Reviewer 1. I agree with the reviewer "... needs to be substantiated with additional parameters in order for any mechanistic insight to be offered. A basic analysis with PICRUSt is not sufficient (and was very poorly developed and integrated within the current manuscript). Directly linking the microbiota with immunological parameters and microbial metabolites would at least allow for some mechanistic associations." I also felt that only basic analysis on the microbiome data is performed. The main result presented in Figure 2 seems not striking. The difference shown in Figure 4 shows only sufficient power within the cohort that has asthmatic mother which do not represent the majority of the population. More in-depth analysis with hopefully more interesting or significant findings would be desirable.

Review 2. Mainly positive. The main critique is on statistical analyses: "It is fundamental for the interpretation of the study results to disclose if breastfeeding practices, hospitalisation, maternal as well as infant antibiotic use and birth mode differed between these two groups". I felt that the authors has done a reasonable job address this issue.

Reviewer 3. A main concern is "the low number of children belonging to the immature PAM cluster 1 at 1 year". I think the authors have tried their best to address the issue given the constraint in their samples.

Another main issue is about the comparison with the CHILD study: "this comparison is important and should be expanded to address additional points. The CHILD study sees different trends in differentially abundant genera, particularly with respect to Veillonella". The seemingly inconsistent results is a concern.

Again, the authors took the critique seriously and provided detailed explanation. It seems to me an plausible explanation is that the results obtained from the two studies is not strong enough and not directly comparable, perhaps due to the limited sample size or the challenge of analyzing complex data.

From my reading, I felt that the analysis results presented in the paper is minimal, it does support their findings. But without more in-depth and/or broader analyses, I am not sure how much does the findings can indicate clinically. It would be desirable to put the results in context better. For example, compared to the diagnosis of another disease or condition. Another way to make the finding more solid is to conduct some sort of replication or validation study, either in another

cohort or use existing data elsewhere.

Reviewers' comments:

Reviewer #1 (neonate gut microbiome, asthma)(Remarks to the Author):

This version of the manuscript is improved and maintains the core message that a certain 'immature' microbiome can predispose children, born from an asthmatic mother, to asthma at 5 years. This is the novel contribution the report brings to the field.

The authors should include a short, referenced, discussion that compares the effect size underlying their primary conclusions (e.g. maternal asthma-immature microbiota-asthma at 5y) with other asthma risk factors, e.g. environment, antibiotics, recurrent viral infections. It is important that this comparison is discussed in order for readers to be able to put the significance of the current report into context.

Response 1: Not many studies provide the predictive strength of an environmental exposure, usually the predictive value of an exposure, though being significantly associated with asthma, is very low. We have now included a new model, which calculates the Area Under the Curve for prediction of asthma based on the microbial composition at age 1-year and risk of asthma at 5 in children born to asthmatic mothers. Please refer to response 7 for details. We have discussed this in relation to the predictive power of genetics, which represents a major risk factor for childhood asthma. We demonstrate a much higher prediction from the microbial composition:

P. 19, line 352: "The univariate genus abundance results were further expanded in a multivariate model, again pointing toward differences in the entire bacterial community structure. For children born to asthmatic mothers, we demonstrated a cross-validated AUC of 0.76 for predicting asthma at age 5 years from the microbial composition at 1-year. In comparison, a large study using genetic risk scores of the highest-ranked asthma risk alleles, demonstrated an AUC of 0.54-0.57 for prediction of school age asthma, and an AUC of 0.59 for preschool persistent wheeze⁴⁸."

Minor comment. Table 1, which is the PICRUSt dataset, has not been removed from the submitted files.

Response 2: Corrected, thank you.

Reviewer #2 (gut microbiome)(Remarks to the Author):

The manuscript submitted by Stockholm et al. describes the microbiota at several time-points in early life in relation to asthma development. The study is conducted within the context of the longitudinal COPSAC2010 birth cohort, making it one of the few studies that moves beyond cross-sectional analyses. Moreover, in this revised version of the manuscript the authors have taken into account the relevant metadata that could have potentially confounded the results. Also other comments by all three previous reviewers have in my opinion in general be carefully addressed. There are only a few issues that still remain that I think should be addressed:

1. Median sequencing depth is insufficient to provide insight in the quality of the data. If the variation in sequencing depth is large, this could have limited the power of the study to reveal additional associations in particular regarding alpha diversity. This aspect is not solved by testing for differences in sequencing depth between groups or adjusting for sequencing depth (although this is a relevant addition the authors made to rule out confounding).

Please provide range in sequencing depth along with median.

Response 3: Thank you for the suggestion. We have now included median and IQR for the descriptions of sequencing depth:

P. 8, line 128: *“With a median sequencing depth of 44,827 reads [IQR: 2,358-78,208] increasing with age of sample”*

P. 14, line 270: *“Furthermore, we found no significant differences in sequencing depth at 1-year between children, who did or did not develop asthma (median [IQR]; asthma vs. non-asthma, 54,260 [12,018-74,971] vs. 51,940 [12,010-73,050], P=0.8); adjustment of the results for sequencing depth resulted in essentially no difference.”*

2. Although the authors have now incorporated the potential confounders in the analyses related to microbial population structure and PAM clustering, it is unclear why the authors chose not to do the same for the alpha-diversity analyses. Associations can also be masked due to confounding, so a lack of association is not an argument not to adjust for covariables. I would recommend the minor effort to adjust alpha diversity analyses as well.

Response 4: Thank you for the suggestion. We have now added this to the alpha-diversity results:

P. 10, line 175: *“There were no significant associations between α -diversity (Shannon and Chao1 indices) at any time-point and asthma risk, also after adjustment for potential confounders (older siblings, duration of exclusive breastfeeding, hospitalization after birth, antibiotic use, and delivery mode).”*

3. The authors indicated in their response that they deliberately focused the manuscript on the entire community structure. However, the point is that the weighted Unifrac that has been used as the only metric for describing population structure and PAM clustering etc., is heavily biased towards the dominant bacterial taxa (see PMID 22711789). By only focussing on this metric as a beta-diversity indice, the aim to explore the entire community structure in association to asthma development is therefore not fully explored. I would recommend to either confirm or extend major findings with an additional indice that is not skewed towards the dominant taxa or address the limitations of using the weighted Unifrac only in the discussion of the manuscript.

Response 5: Thank you for the suggestion. We have now included other beta-diversity measures as sensitivity analyses:

P. 11, line 192: *“In a test of sensitivity, we also examined associations between population structure at age 1-year in children born by asthmatic mothers and asthma risk using other β -diversity indices that are not biased by the dominant taxa (unweighted UniFrac and Jensen–Shannon divergence), which yielded similar results (P<0.001).”*

Reviewer #3 (human microbiome)(Remarks to the Author):

Authors have attempted to respond to first round of reviews
My concerns remain - incremental advance, 16S only data set of limited value and follow up samples are critical

Response 6: We recognize that our study is observational and that no conclusions about causality and

directionality can be reached, which is already stated thoroughly in the manuscript. Nevertheless, despite this important limitation, we believe this work represents a substantial advance in the understanding of asthma development in childhood.

Reviewer #4 (functional genomics, biostat)(Remarks to the Author):

In this paper, Stokholm and colleagues characterized the microbial composition of fecal samples from children in the COPSAC child cohort at three different ages (1 week, 1 month, and 1 year) using 16S rRNA gene sequencing. The goal of this prospective study was to evaluate the association between the gut microbiota during the first year of life and asthma diagnosis in later year. The current manuscript has been revised from the original submission. The authors have addressed every issue raised by the three previous reviewers. I am in agreement with most of the critiques raised by the three previous reviewers. My review is primarily focused on whether the responses from the authors are adequate and the revised version has been improved. I only have minor points raised from my own reading of the manuscript. Overall, I felt that the cohort is a unique with both microbiome data as well as deep clinical information and the study is well designed to take advantage of the prospective cohort with sufficient power. The finding is interesting and make sense. However, given the current analysis result, the difference appears to be weak, which cast doubt whether the finding would have any clinical implication.

Reviewer 1. I agree with the reviewer "... needs to be substantiated with additional parameters in order for any mechanistic insight to be offered. A basic analysis with PICRUSt is not sufficient (and was very poorly developed and integrated within the current manuscript). Directly linking the microbiota with immunological parameters and microbial metabolites would at least allow for some mechanistic associations." I also felt that only basic analysis on the microbiome data is performed. The main result presented in Figure 2 seems not striking. The difference shown in Figure 4 shows only sufficient power within the cohort that has asthmatic mother which do not represent the majority of the population. More in-depth analysis with hopefully more interesting or significant findings would be desirable.

Response 7: We thank the reviewer for the work put into reviewing the manuscript. We have already removed the PICRUSt analysis in the previous revision. Furthermore, it is true that the estimates at the population level are lower compared with the more specific phenotypes of children born to mothers with asthma. However, we still observe strong risk estimates (All: PAM HR: 2.87; MAZ HR: 1.77) and especially if the child is born to an asthmatic mother (PAM HR: 12.99; MAZ HR: 6.53). In our opinion, this very specific phenotype that seems susceptible to microbial impacts strengthens the findings. This may be of special significance since with confirmation by adequate experimental models, future interventions involving microbial manipulation could target this particular group of children. Further corroborating the association is the finding that microbial maturity associates with protection from an allergic asthmatic phenotype that persists to age 5.

In this revision, we now include a cross-validated sparse PLS model between microbial genus abundance at age 1-year and asthma at age 5 in children born to asthmatic mothers. This supervised model was constructed to identify jointly contributing taxa. This resulted in a one-component model based on the relative abundances of 60 genera with high predictive capacity for asthma (cross-validated AUC 0.76). Again, this suggests that a possible causal effect is not mediated by a single or by few taxa, but rather by a more global change in composition. Please refer to **Fig. S6**.

Methods:

P. 24, line 463: *"We used sparse Partial Least Squares (PLS)⁶⁸ modeling of asthma at age 5-years after filtering genera (prevalence >5% of children, >0.01% mean relative abundance) and log-transforming relative abundances, using half the lowest nonzero value as a pseudocount. We selected the optimum*

number of input variables using repeated 10-fold cross-validation of the Area Under the Curve (AUC) statistic to avoid overfitting. The final model was chosen by the highest median AUC value. The predicted values of left out folds were combined to a PLS score.”

Results:

P. 12, line 215: “To further examine the associations between the microbial composition at age 1-year and asthma at age 5 in children born to asthmatic mothers, a cross-validated sparse PLS model was constructed to identify jointly contributing taxa at age 1-year predicting asthma in these children. This resulted in a one-component model based on the relative abundances of 60 genera (Fig. S6). The model demonstrated a high predictive capacity for asthma (cross-validated AUC 0.76) and the many contributing taxa suggest an overall delayed microbial maturation in these children at age 1-year.”

Discussion:

P. 19, line 352: “The univariate genus abundance results were further expanded in a multivariate model, again pointing toward differences in the entire bacterial community structure. For children born to asthmatic mothers, we demonstrated a cross-validated AUC of 0.76 for predicting asthma at age 5 years from the microbial composition at 1-year. In comparison, a large study using genetic risk scores of the highest-ranked asthma risk alleles, demonstrated an AUC of 0.54-0.57 for prediction of school age asthma, and an AUC of 0.59 for preschool persistent wheeze⁴⁸.”

P. 20, line 377: “Because our results are observational and most of the taxa associated with lower asthma risk demonstrate collinearity, we are not able to pinpoint specific causal taxa; however, the multivariate model illustrates many jointly contributing taxa and not a single risk taxon, consistent with a global effect on the microbiota.”

Review 2. Mainly positive. The main critique is on statistical analyses: “It is fundamental for the interpretation of the study results to disclose if breastfeeding practices, hospitalisation, maternal as well as infant antibiotic use and birth mode differed between these two groups”. I felt that the authors has done a reasonable job address this issue.

Response 8: Thank you.

Reviewer 3. A main concern is “the low number of children belonging to the immature PAM cluster 1 at 1 year”. I think the authors have tried their best to address the issue given the constraint in their samples.

Response 9: Thank you. The low number of children belonging to the immature PAM cluster 1 at 1 year was a consideration for us as well. As such, we performed the additional maturity analysis by MAZ, resulting in equal numbers in the 2 groups (high vs. low maturity), which supported the findings from PAM-clustering.

Another main issue is about the comparison with the CHILD study: “this comparison is important and should be expanded to address additional points. The CHILD study sees different trends in differentially abundant genera, particularly with respect to Veillonella”. The seemingly inconsistent results is a concern.

Again, the authors took the critique seriously and provided detailed explanation. It seems to me an plausible explanation is that the results obtained from the two studies is not strong enough and not directly comparable, perhaps due to the limited sample size or the challenge of analyzing complex data.

From my reading, I felt that the analysis results presented in the paper is minimal, it does support their

findings. But without more in-depth and/or broader analyses, I am not sure how much does the findings can indicate clinically. It would be desirable to put the results in context better. For example, compared to the diagnosis of another disease or condition. Another way to make the finding more solid is to conduct some sort of replication or validation study, either in another cohort or use existing data elsewhere.

Response 10: We have now added additional analyses to the manuscript. Please refer to responses 4, 5 and 7. In our opinion, the observations that microbial maturity associates with protection from an allergic asthma phenotype that persists to age 5, as well as the number of asthmatic episodes in childhood further strengthens the results. To our knowledge, no other cohorts exist with both robust asthma phenotypes diagnosed in the clinic throughout childhood and longitudinally collected samples before the onset of disease that have been characterized by microbial sequence analysis.

Reviewers' Comments:

Reviewer #1 (Remarks to the Author):

The authors have sufficiently addressed my concerns.

Reviewer #2 (Remarks to the Author):

The authors have carefully addressed the remaining comments. I have no further comments.

Reviewer #4 (Remarks to the Author):

I felt that the authors have adequately addressed all the issues I have raised in the previous round of review.